# Establishment of Two Novel Ovarian Tumor Cell Lines with Characteristics of Mucinous Borderline Tumors or Dedifferentiated Carcinoma—Implications for Tumor Heterogeneity and the Complex Carcinogenesis of Mucinous Tumors

**DOI:** 10.3390/cancers17101716

**Published:** 2025-05-20

**Authors:** Hasibul Islam Sohel, Umme Farzana Zahan, Tohru Kiyono, Masako Ishikawa, Sultana Razia, Kosuke Kanno, Hitomi Yamashita, Shahataj Begum Sonia, Kentaro Nakayama, Satoru Kyo

**Affiliations:** 1Department of Obstetrics and Gynecology, Faculty of Medicine, Shimane University, Izumo 693-8501, Japan; hasibulsohel1167@gmail.com (H.I.S.); farzanashormi99@gmail.com (U.F.Z.); m-ishi@med.shimane-u.ac.jp (M.I.); kanno39@med.shimane-u.ac.jp (K.K.); meme1103@med.shimane-u.ac.jp (H.Y.); sbsonia1995@gmail.com (S.B.S.); 2Exploratory Oncology Research and Clinical Trial Center (EPOC), National Cancer Center, Kashiwa 277-8577, Japan; tkiyono@east.ncc.go.jp; 3Department of Legal Medicine, Faculty of Medicine, Shimane University, Izumo 693-8501, Japan; raeedahmed@yahoo.com; 4Department of Obstetrics and Gynecology, East Medical Center, Nagoya City University, Nagoya 464-8547, Japan

**Keywords:** ovarian tumor, mucinous borderline ovarian tumor, mucinous carcinoma, dedifferentiated carcinoma

## Abstract

Mucinous borderline tumors of the ovary (MBOTs) are characterized by intermediate malignant potential; however, they are known to occasionally recur. The molecular mechanisms driving their carcinogenesis and tumor biology remain poorly understood, and experimental models for this tumor type are scarce. In the study presented herein, we established two independent cell lines from the tissues of a patient with an MBOT. In a patient-derived xenograft model, HMucBOT-1 retained MBOT morphology, whereas HMucBOT-2 exhibited features of mucinous carcinoma with undifferentiated components, with unique genetic alterations, including *KRAS* amplification. These cell lines can be used as a preclinical model of MBOTs and serve as a valuable tool for studying tumor heterogeneity and genetic diversity that may induce treatment failure or relapse.

## 1. Introduction

Cancer continues to represent a significant global health burden, accounting for millions of deaths each year. In 2022, an estimated 20 million new cancer cases and 9.7 million cancer-related deaths were reported worldwide [1]. In the United States, approximately 2.04 million new cancer cases and 618,000 deaths are projected for 2025 [2]. Among these, ovarian cancer remains one of the most lethal gynecologic malignancies, responsible for an estimated 206,839 deaths globally and 12,730 deaths annually in the United States [1,2]. Borderline ovarian tumors (BOTs), sometimes called tumors of low malignant potential, constitute a significant portion of ovarian cancer, accounting for 10–15% of all epithelial ovarian malignancies and a 5-year survival rate of 98% [3,4]. BOTs are fundamentally characterized by intermediate clinical aggressiveness with pathological features of moderate cellular proliferation and slight nuclear atypia but lacking destructive stromal invasion [5,6,7]. BOTs are classified into six subtypes based on the type of epithelial cells present as follows: serous (50%), mucinous (45%), and the less common endometrioid, clear cell, seromucinous, and borderline Brenner tumor [6,8]. Mucinous borderline tumors (MBOTs) and serous borderline tumors (SBOTs) account for over 90% of all BOTs [9], with MBOTs being the second most common type of BOT worldwide [10]. In Japan, MBOTs are the most common type of ovarian tumor, accounting for 57.7% of all cases, followed by SBOTs at 20.4% [11]. MBOTs, also known as atypical proliferative mucinous tumors, feature mucin-producing cells with borderline malignant potential [12,13,14]. The five-year survival rate for MBOTs without intraepithelial carcinoma or micro-invasion is approximately 95–98%, with a 1% mortality and a 4.2–7% recurrence risk, some of which may progress to MOCs [15,16,17,18,19,20]. MBOT is characterized by specific gene mutations of *ARID1A*, *ARID1B*, *KRAS*, *CDKN2A*, *PIK3CA*, *TP53*, *PTEN*, *GNA11*, or *ERBB2*, and almost the same mutations are also seen in MOC [21,22].

The most representative mutations in mucinous tumors are *KRAS* mutations and ERBB2 amplification/overexpression, which activate the MAPK signaling pathway, thereby promoting cell proliferation and tumor progression [23,24]. Wakazono et al. demonstrated that whole-genome doubling and extensive copy number alterations, including amplification of the *KRAS* mutant allele, were specifically observed in MOC recurrences arising from MBOTs [20]. In particular, while *KRAS* mutations are not observed in healthy ovaries, they are detected in 57% of benign mucinous ovarian tumors, 90% of MBOTs, and 76% of MOCs [25]. These findings suggest that *KRAS* mutations and copy number alterations may play a crucial role in the development of mucinous carcinoma.

Recent advances in cancer therapy have shifted from conventional approaches to more precise, personalized treatments driven by molecular profiling and targeted strategies [26]. However, the lack of robust preclinical models for rare tumors like MBOTs remains a barrier to understanding their pathogenesis and designing effective treatment strategies.

In the current study, we present two newly established novel cell lines derived from the tissues of a patient with an MBOT that are expected to aid in understanding the molecular carcinogenesis and heterogeneity of this rare tumor type, with the potential to lead to the identification of novel therapeutic targets.

## 2. Methods and Materials

### 2.1. Cell Preparation and Establishment of the Cell Lines

This study was approved by the Ethics Review Board of Shimane Medical University (IRB No. 20070305-1 and 20070305-2). The cell lines were established using a surgically removed ovarian tumor sample from a 62-year-old patient. The operation was performed at Shimane University Hospital, and the patient provided written informed consent for the use of her clinical and pathological specimens in this study.

Tissue samples were collected and submitted to primary culture based on previously established protocols [27,28]. In detail, tissues were collected from two different sites of the ovarian tumor’s papillary structure and then ground separately using sterile scissors, followed by gentle washing with sterile phosphate-buffered saline (PBS) to remove blood contaminants. Thereafter, the tissue fragments were enzymatically digested with collagenase type III (Washington Biochemical Corp., Lakewood, NJ, USA) supplemented with a ROCK inhibitor (Selleck Chemicals, Houston, TX, USA) and subsequently washed to remove residual debris. The digested tissues were then resuspended in 5% Dulbecco’s modified Eagle’s medium (DMEM) (WAKO, Osaka, Japan) and incubated with gentle agitation at 25 °C for 48 h to promote epithelial cell detachment. Lastly, purified epithelial cells were subjected to two independent primary cultures derived from different sites of the ovarian tumor.

To extend the life span of primary cultured cells, human telomerase reverse transcriptase (hTERT), Cyclin D1, and CDK4 (CDK4R24C: an inhibitor-resistant variant of CDK4) were overexpressed through lentivirus-mediated gene transfer through the utilization of the previously published gateway method [29,30]. The resultant cells were designated as HMucBOT-1 and HMucBOT-2, cultured in a mixture of DMEM and F12 (Thermo Fisher Scientific Inc., Waltham, MA, USA) at a ratio of 1:3 (WAKO, Osaka, Japan), supplemented with 5% fetal bovine serum (FBS; Gibco, Carlsbad, CA, USA), 1% penicillin–streptomycin (Pen-Strep; Sigma-Aldrich, St. Louis, MO, USA), Adenine HCL (WAKO, Osaka, Japan), Insulin (Sigma-Aldrich, St. Louis, MO, USA), Y-27632.2HCL (Selleck Chemicals, Houston, TX, USA), 17β-estradiol (Sigma-Aldrich, St. Louis, MO, USA), Hydrocortisone (Sigma-Aldrich, St. Louis, MO, USA), hEGF (PEPROTECH, Rocky Hill, NJ, USA), DMH1 (R&D Systems, Minneapolis, MN, USA), A-83-01 (R&D Systems, Minneapolis, MN, USA), and Cholera toxin (WAKO, Osaka, Japan), and maintained in an incubator with 5% CO_2_ at 37 °C [28].

### 2.2. Short Tandem Repeat (STR) Analysis

Genomic DNA was extracted from the original tumor tissue and its derived cultured cells (HMucBOT-1 and HMucBOT-2) using the DNeasy Blood and Tissue Kit (Qiagen, Hilden, Germany). DNA concentrations were measured using a NanoDrop 8000 spectrophotometer (Thermo Fisher Scientific, Waltham, MA, USA), and the samples were stored at −20 °C until further use. The samples were then submitted for STR profiling analysis (BEX Co., Ltd., Tokyo, Japan) targeting 16 loci (Appendix A) [31]. The data generated were interpreted using either GeneMapper 6.0 (Applied Biosystems, Thermo Fisher Scientific, Waltham, MA, USA) or PeakScanner software v1.0 (Applied Biosystems, Thermo Fisher, Waltham, MA, USA). The obtained STR profiles were compared with reference profiles from the ACC, DSMZ, and JRCB databases to assess matching.

### 2.3. Mouse Tumorigenicity

Female C.B-17/Icr-scid/scidJc1 (CLEA Japan, Inc., Shizuoka, Japan) SCID mice (mice were acclimatized for 4 weeks before experiments) aged 5–7 weeks were injected intraperitoneally and subcutaneously on the right dorsum with different amounts of the two cell lines (2 × 10^4^ or 2 × 10^6^ cells). Tumor growth was monitored through regular inspection and palpation twice a week. The tumors were then fixed in 4% buffered formaldehyde, and sections were stained with hematoxylin and eosin for further analysis.

### 2.4. Whole-Exome Sequencing

Whole-exome sequencing analysis was performed between the original tumor samples and the HMucBOT-2 cell line to compare the genetic alterations between them; the techniques utilized for genome sequencing are discussed in a previous publication [20]. The Agilent 2000 Tape Station (Agilent Technologies, Santa Clara, CA, USA) was first used to assess DNA integrity. Subsequently, Illumina MiSeq (San Diego, CA, USA) whole-exome sequencing using enriched amplicons was performed. The sequencing data were analyzed using the Genome Jack bioinformatics pipeline (Mitsubishi Space Software Co., Ltd., Tokyo, Japan). High analytical sensitivity and specificity were ensured throughout the investigation through the use of sequence alignment, variant calling, variant filtering, variant annotation, and variant prioritization.

### 2.5. Immunohistochemistry Assay

Immunohistochemical analysis was carried out on the paraffin-embedded tissues of mouse xenograft tumors. Briefly, 5 µm thick sections of paraffin-embedded tissues were serially cut. Selected sections were used for immunohistochemical (IHC) analysis or stained with hematoxylin and eosin for histological analysis. For the immunohistochemical analysis, the sections were deparaffinized and incubated with antibodies of CK7, Pax8, MUC1, AE1/AE2, CAM5.2, Desmin, EMA, S-100, Twist, Snail, Vimentin, and DMMR proteins overnight at 4 °C (Appendix A). Tris and EDTA buffer (pH 9, Ref-S3467, Dako, Carpinteria, CA, USA) was used for antigen retrieval. Using a light microscope, the samples were evaluated by pathologists at the department of pathology in our hospital, who were blinded to the clinicopathological variables.

### 2.6. Cell Proliferation Assay

The HMucBOT-1 and HMucBOT-2 cells were plated in U-bottomed 96-well plates at a density of 4000 cells per well. Cell count was measured indirectly using a methyl thiazolyl tetrazolium (MTT) assay [28,32]. The results are expressed as the mean ± standard deviation (SD) from independent triplicate experiments.

### 2.7. Soft Agar Colony Formation Assay

A total of 10,000 HMucBOT-1 and HMucBOT-2 cells were plated into 24-well plates with a top layer of 0.33% noble agar in 2X DMEM with 5% FBS and a bottom layer of 0.5% base agar in 2X DMEM with 5% FBS. Once the gel had solidified, each well was covered with 1 mL of culture medium and incubated at 37 °C. After 2 weeks, colonies larger than 0.05 mm in diameter were counted.

### 2.8. Invasion Assay

A cell invasion assay was conducted on the HMucBOT-1 and HMucBOT-2 cell lines using a Boyden chamber equipped with filter inserts (pore size: 8 µm), which were coated with Matrigel (40 µg; Collaborative Biomedical, Becton Dickinson Labware, Bedford, MA, USA) and placed in 24-well plates (Nucleopore, Pleasanton, CA, USA). In the assay, 2500 cells were seeded in the upper chamber, and the same medium was added to the lower chamber. After incubation for 24 h, the cells were fixed with methanol and stained with 0.05% crystal violet in phosphate-buffered saline (PBS) for 1 h at room temperature. Cells on the upper side of the filters were removed using cotton-tipped swabs, and the filters were rinsed with PBS. The invasive cells on the lower side of the filters were counted in four different fields at 200× magnification.

### 2.9. Migration Assay

The cell migration assay was conducted using wound-healing assays. HMucBOT-1 and HMucBOT-2 cells were plated in 6-well plates and allowed to adhere overnight. A sterile 200 μL pipette tip was used to create a scratch in the cell monolayer, generating a wound. The cells were then incubated for 24 h. Cells that moved into the wound area were considered migrating cells and were observed using an inverted Olympus phase-contrast microscope (Olympus Taiwan, Taipei, Taiwan). The healing rate was assessed by counting the number of migrating cells in four different areas for each assay.

### 2.10. Responses to Chemotherapeutic Agents

HMucBOT-1 and HMucBOT-2 cells (2 × 10^3^ per well) were seeded in triplicate into flat-bottomed, 96-well microtiter plates and incubated for 24, 48, and 72 h with culture media. The cells were then treated with paclitaxel at concentrations of 5, 10, 30, or 20 nmol/L or cisplatin at concentrations of 5, 15, 45, or 135 µmol/L. Cells treated with dimethyl sulfoxide (DMSO) at the corresponding concentrations served as negative controls. Next, 30 µL of 3-(4,5-dimethylthiazol-2-yl)-2,5-diphenyltetrazolium bromide solution (5 mg/mL) was added, and the mixture was incubated for 4 h. Thereafter, 100 µL of DMSO was added to dissolve the formazan crystals, and the mixture was shaken vigorously for 15 min. The optical density at 490 nm was measured using an enzyme-linked immunosorbent assay reader. Lastly, the IC50 value was calculated and evaluated using GraphPad Prism software (Version 7.04, GraphPad Software, LLC, San Diego, CA, USA).

### 2.11. Statistical Analysis

Data were analyzed using SPSS version 27 (SPSS Inc., Chicago, IL, USA). For continuous variables, Student’s *t*-test was applied, and Fisher’s exact test was used for categorical variables. Results from the three separate experiments are reported as the mean ± standard error.

## 3. Results

### 3.1. Histopathological Overview of the Original Tumor

In the present study, we focused on a 62-year-old woman suffering from a large ovarian cystic tumor that was considered benign to borderline upon preoperative diagnosis, given the size of the tumor and the presence of abundant partial-thickness septa but lacking apparent solid component within the tumor (Figure 1A). The patient underwent a bilateral salpingo-oophorectomy, revealing a right ovarian tumor measuring 24 × 18 cm. The post-operative pathological evaluation revealed that the tumor had a cystic structure lined with mucinous cells. Most of the tumor showed features typical of mucinous cystadenoma, including changes in the cyst wall, such as hyalinization and edema. In a few areas, papillary structures with cellular atypia were present, such as an increased nucleus-to-cytoplasm ratio and prominent nucleoli, but without stromal invasion. These features suggest a mucinous borderline malignancy (Figure 1B). Based on the above finding, the final pathological diagnosis was determined to be an MBOT. At present, the patient is being monitored closely without adjuvant chemotherapy.

### 3.2. Establishment of the MBOT Cell Lines

Tissues were collected from two different sites of the ovarian tumor’s papillary structure shortly after the operation, and epithelial cells were isolated, purified, and subjected to primary culture (Figure 2). These cells were confirmed to exhibit typical epithelial morphology. To enable in-depth research, we sought to extend the life span of these cells, as isolated cells from clinical tumors are likely to undergo senescence once subjected to primary culture. There are two types of senescence: premature senescence and telomere-dependent senescence. The former can be overcome by inhibiting Rb activation, possibly through overexpression of activated CDK4 (CDK4R24C) and Cyclin D1, and the latter by overexpressing human telomerase reverse transcriptase (hTERT) to induce telomerase activity [33,34,35]. We therefore overexpressed these three factors through lentiviral transfection and successfully obtained cells with extended life spans. These cells continued to proliferate until at least 60 population doublings (PDs), indicating that they can be stably cultured for use in subsequent analyses.

The resulting cell lines were designated as HMucBOT-1 and HMucBOT-2, both exhibiting a polygonal, tightly packed morphology, characteristics of epithelial cells with Pan cytokeratin expression, resembling the cobblestone pattern (Figure 2). The STR patterns observed did not match any previously reported cell lines (Appendix A), confirming that HMucBOT-1 and HMucBOT-2 are original novel cell lines.

Epithelial fractions of sampled tissues were subjected to primary culture, followed by overexpression of *CyclinD1*, *CDK4*, and *hTERT* to overcome cellular senescence during the course of primary culture. Over time, these cells showed immortal features, preserving epithelial characteristics, such as Pan-cytokeratin (CK) expression.

### 3.3. Histological Diversity of Mouse Tumors Derived from HMucBOT-1 and HMucBOT-2 Cells

Although HMucBOT-1 and HMucBOT-2 cells were derived from the ovarian tumor pathologically diagnosed as an MBOT, there was no guarantee that these cells specifically originated from borderline lesions because the areas showing borderline histology were limited within the tumor in histological sections, which could not be macroscopically identified. We therefore sought to confirm their histological characteristics and malignant potential based on their tumorigenicity in mice.

The cells were independently inoculated into the SCID mice, and we were able to efficiently determine tumor development within 15–17 weeks following the injection of 2 × 10^4^ cells or 12–14 weeks after the injection of 2 × 10^6^ cells (Table 1).

Histological analysis involving HE staining of the tumor sections showed distinct tumor characteristics. Tumors derived from HMucBOT-1 cells retained the MBOT histology (Figure 3), consistent with the primary tumor’s features. However, unexpectedly, tumors derived from HMucBOT-2 cells exhibited carcinoma histology, with a mixture of mucinous and high-grade-type undifferentiated carcinoma components. We pathologically diagnosed this mouse tumor as dedifferentiated carcinoma. Based on the above results, it can be concluded that the histology of HMucBOT-1 and HMucBOT-2 cells significantly diverged in the mouse tumor model, possibly reflecting the heterogeneity of the original tumor.

HMucBOT-1 and HMucBOT-2 cells were injected into SCID mice, and tumorigenicity was observed. Both cells efficiently formed tumors, and HE staining of the tumors of HMucBOT-1 cells showed mucinous borderline histology, consistent with the original tumor. However, tumors of HMucBOT-2 cells exhibited dedifferentiated carcinoma, composed of mucinous carcinoma and undifferentiated carcinoma lesions. White arrows, mucinous carcinoma lesions; Black arrows, undifferentiated carcinoma lesions.

### 3.4. HMucBOT-2 Cells Showed Genetic Alterations Distinct from the Original Tumor

We focused on the findings that HMucBOT-2 cells showed undifferentiated carcinoma histology on mouse tumors despite the fact that the original tumors were pathologically diagnosed as MBOTs. To examine the molecular backgrounds of HMucBOT-2 cells, we performed whole-exome sequencing on macroscopically dissected MBOT lesions from the HE sections of the original tumors in addition to the HMucBOT-2 cells. Oncogenic mutations in *KRAS*, *ARID1A*, *ARID1B*, *SMO*, *NF1*, and *BRIP1* were frequently detected in both samples (Table 2 and Table 3).

No threefold or higher copy number alterations were observed in the original tissue (Figure 4A). However, in HMucBOT-2 cells, significant chromosomal copy number alterations were identified in chromosome (Chr) 12, with it displaying a threefold gain in *KRAS*, *KMT2D*, *CLIP1*, and *CDK4* (Figure 4B), and Chr 5, with it displaying a threefold gain in *TERT*, with some of these changes possibly being due to artificial genetic manipulations via CDK4 and hTERT overexpression. Copy number alterations were also observed in other chromosomes; however, they did not reach the threefold threshold.

### 3.5. Biological Behavior of HMucBOT-1 and HMucBOT-2 Cells

We were interested in investigating the differences between the biological behavior of both cells. To help us achieve this aim, we first tested the growth rates of the HMucBOT-1 and HMucBOT-2 cells using an MTT assay. The HMucBOT-2 cells showed a much greater proliferation capacity compared to the HMucBOT-1 cells, with the doubling time for the HMucBOT-2 cells being 58 h compared to 87 h for the HMucBOT-1 cells (Figure 5A) [36,37]. Their ability to form colonies on soft agar was subsequently assessed, and the HMucBOT-2 cells exhibited significantly greater colony-forming potential than the HMucBOT-1 cells (Figure 5B). Furthermore, the migration and invasion assay results demonstrated that HMucBOT-2 cells exhibited a higher capacity for both invasion and migration compared to HMucBOT-1 cells (Figure 5C,D). These findings highlight how HMucBOT-2 cells exhibit more aggressive and invasive features than HMucBOT-1 cells. Consistent with previous reports, our results indicate that *KRAS* amplification and additional copy number alterations in genes may confer greater tumor aggressiveness compared to *KRAS* mutation alone [38,39,40].

### 3.6. Characterization of Mouse Xenograft Tumors

To further characterize xenograft tumors of the HMucBOT-1 and HMucBOT-2 cells, we performed an immunohistochemical analysis. Typical epithelial markers were first examined in both cell lines and tumors. The HMucBOT-1 tumor expressed the epithelial markers CK7, PAX8, and MUC1 in the borderline lesions (Figure 6A); in addition, these markers were expressed in cell lines (Appendix A). In a similar manner, in the HMucBOT-2 tumor, mucinous carcinoma lesions expressed all of these markers, with undifferentiated lesions exhibiting PAX8 expression but not CK7 or MUC1 expression.

We next focused on the expression of the dedifferentiated markers for the HMucBOT-2 tumor based on the results of previous studies [41,42,43,44,45]. The HMucBOT-2 tumor exhibited AE1/AE3, CAM 5.2, and EMA expression in mucinous carcinoma lesions but not undifferentiated lesions (Figure 6B). No significant expression of Desmin or S-100 was detected in both mucinous and undifferentiated carcinoma, consistent with the expression pattern of dedifferentiated carcinoma.

We further extended the IHC analysis for the HMucBOT-2 tumor in relation to epithelial–mesenchymal transition (EMT) markers. Twist expressions were negative for both mucinous and undifferentiated carcinoma lesions, whereas Snail and Vimentin expressions were positive in undifferentiated carcinoma lesions (Figure 6C).

The involvement of mismatch repair deficiency has been noted in dedifferentiated carcinoma in endometrial cancer [46,47,48]. We therefore investigated the marker expression of MLH1, MSH2, MSH6, and PMS2; however, they were all found to be present in both mucinous carcinoma and undifferentiated lesions, indicating the mismatch repair proficiency of this tumor (Figure 6C).

### 3.7. The Response to Cytotoxic Drugs

To further characterize the two types of cells in terms of clinical relevance, we conducted experiments to determine the cells’ responses to cytotoxic agents used in clinical practice to treat ovarian cancer. Because taxanes and platinum-based agents are the only representative chemotherapeutic agents for ovarian mucinous tumors, we examined the sensitivity of paclitaxel and cisplatin. Cell viability did not change significantly when the cells were treated with serial concentrations of paclitaxel or cisplatin for 24 h. However, it significantly decreased when the cells were treated for 72 h, with the half-maximal inhibitory concentration (IC50) of paclitaxel being roughly 11.6 and 12.5 nmol/L for HMucBOT-1 (Figure 7A) and HMucBOT-2, respectively, and the IC50 of cisplatin being 26.6 and 29.3 mmol/L, respectively (Figure 7B). These results indicated that these cells were at least partially sensitive to paclitaxel and cisplatin in vitro.

## 4. Discussion

Studies on ovarian carcinogenesis have primarily focused on ovarian cancer, with much less focus on borderline tumors, because the origin of the latter (from a healthy ovary or a benign cyst?) and their natural history remain largely unknown. In addition, the question of whether borderline tumors are precursors of ovarian cancer has yet to be definitively answered. Borderline tumors are rare tumors, and their clinical, pathological, and molecular characteristics significantly differ among each histological subtype, making research on this tumor type extremely difficult. Understanding the pathogenesis of borderline tumors therefore presents a challenge for gynecologic oncologists.

MBOTs are the second most prevalent type of ovarian tumor, accounting for 35–45% of all ovarian borderline tumors [10]. Although a limited number of articles cover the characteristics and clinical management of MBOTs, including their prognosis and recurrence patterns [19,49,50], the establishment of permanent cell lines from MBOTs remains constrained due to their rarity and the difficulty of stable culture. Patient-derived tumor cell lines retain essential tumor features and offer valuable platforms for investigating tumor biology, drug sensitivity, and personalized therapeutic strategies [51,52].

In the present study, we successfully established two novel cell lines, HMucBOT-1 and HMucBOT-2, from primary cultures derived from the tissues of a patient with an MBOT. The use of CyclinD1/CDK4 together with hTERT overexpression has been recognized as an established method to extend cellular life span, preserving the original nature of primary epithelial cells from a variety of tumor types [33,34,35]. These cells and their derived tumors show positive expression for the CK7, PAX8, and MUC1 proteins, confirming that they are of epithelial origin, possibly from mucinous adenoma or borderline lesion [27,53,54,55,56].

Despite originating from the same tumor, the two cell lines displayed distinct pathological characteristics. HMucBOT-1 retained morphological features of the original MBOT when xenografted into SCID mice, while HMucBOT-2 exhibited dedifferentiated carcinoma with features of both mucinous and high-grade undifferentiated carcinoma. At the time of tissue sampling following surgery on the patient’s tumor, we collected tissues from the papillary thickened sites of the septa; however, there was no guarantee that such areas specifically contained borderline cells because borderline lesions were not macroscopically detected, unlike during microscopic inspection. We speculate that cells with borderline characteristics or with more potential were fortunately included in the collected tissues, which might be the origin or source of HMucBOT-1 or HMucBOT-2 cells, considering their tumorigenic potential in mice.

The emergence of dedifferentiated carcinoma from HMucBOT-2 xenografts is notable, especially considering that histological findings from the original tumor demonstrated MBOTs with only small portions of borderline lesions. We speculate that there are two possible explanations for this diversity: either cells with borderline acquired carcinoma traits during culture or mouse xenografting or cells with carcinoma features were already present in the original tissues but only in minute portions that could not be detected through routine pathological inspection. We favor the latter hypothesis, which may reflect the tumor heterogeneity in MBOTs. In other words, MBOTs may contain cells with more malignant characteristics in a limited range of areas even under the final pathological diagnosis of MBOTs.

The whole-exome sequencing revealed a variety of common oncogenic mutations in the original tumor and HMucBOT-2, including *KRAS*, *ARID1A*, *ARID1B*, and *NF1*, while *CDKN2A* remained wild-type. Although DNA from FFPE and cultured cells can introduce technical biases [57], we followed stringent processing and quality control protocols to minimize these biases. Our consistent mutational profiles across original and derived samples underscore the reliability of our findings. Notably, while *KRAS*, *CDKN2A*, and *ARID1A* mutations are frequently observed in both MBOTs and mucinous ovarian carcinomas [21,58,59], our analysis also identified mutations in *ARID1B*, *SMO*, *BRIP1*, and *NF1*, as well as amplification of *KMT2D* and *CLIP1* genes, which were not previously associated with MBOTs and MOCs, indicating novel molecular events of these tumors.

In particular, HMucBOT-2 demonstrated copy number gains in chromosomes 5 and 12 where *KRAS*, *KMT2D*, and *CLIP1* are located and have been associated with cancer transformation.

*KRAS* mutations are the most frequent alterations observed in MOC (64.9%) and MBOTs (92.3%) [21], suggesting that our cell lines may therefore preserve or reflect the genetic features of these tumors. As a key oncogene, *KRAS* regulates the signaling pathways critical for cell proliferation and survival [60]. Mutations in *KRAS* can lead to constitutive activation of these pathways, resulting in uncontrolled cell growth and tumor development. When the mutant *KRAS* allele undergoes amplification, the increased gene copy number can lead to elevated expression of the mutant protein, further enhancing downstream signaling activity and promoting cancer progression [40,61].

Abnormalities in the *KMT2D* (lysine methyltransferase 2D) gene contribute to the development of various types of cancer, including medulloblastoma, melanoma, lymphomas, leukemia, and cancers of the lung, prostate, kidney, bladder, ovary, pancreas, esophagus, and stomach [62,63,64,65,66,67,68,69,70,71,72,73,74,75,76], disrupting genetic and epigenetic regulation and promoting tumor initiation, progression, and metastasis.

The *CLIP1* (CAP-Gly-domain-containing linker protein 1) gene regulates microtubule dynamics and mediates cell migration, particularly in cancer metastasis. It aids the transport of proteins and organelles along microtubules, contributing to tumor progression in several types of cancer [77].

In fact, the results of the present in vitro study demonstrated that HMucBOT-2 cells show greater colony formation, invasion, and migration compared to HMucBOT-1 cells. Based on these findings, we speculate that additional genetic alterations observed in HMucBOT-2 cells may be involved in the development of dedifferentiated carcinoma and contribute to more aggressive behavior.

Although MBOTs rarely progress to invasive carcinoma, such progression appears to be driven by genomic events like mutant *KRAS* amplification and widespread copy number alterations. Amplified *KRAS* intensifies Ras–MAPK signaling, promoting uncontrolled proliferation [24,78]. Meanwhile, high copy number alterations in genes are the key drivers that influence the grade and progression of metastatic disease, contributing to uncontrolled cell growth, increased invasion, and therapy resistance [79,80,81]. Several previous studies have reported that high copy number alterations, along with amplification of mutant *KRAS* alleles and/or co-occurrence with *TP53* mutations, may contribute to the progression of MBOTs to MOCs [20,82,83,84,85,86].

Recent advances in non-invasive diagnostics may facilitate early detection of these transitions. Liquid-biopsy-based detection of circulating tumor DNA (ctDNA) allows real-time monitoring of tumor-derived mutations [87]. Molecular barcoding has further improved the sensitivity of sequencing, allowing for the detection of low-frequency variants that may be missed by conventional methods [88]. Additionally, cell-free DNA (cfDNA) methylation profiling can reveal cancer-specific patterns for early diagnosis [89]. Together, these technologies may support individualized surveillance strategies and reduce misdiagnosis by improving molecular accuracy. Shishi He and colleagues established cell lines from both benign and malignant breast phyllodes tumors (PTs) obtained from different patients. These biphasic tumors contain both epithelial and stromal components. The established cell lines were xenografted into mice and subjected to various in vitro analyses. Their study provides a valuable model for exploring tumor heterogeneity and identifying potential therapeutic targets or biomarkers [90]. This approach supports our study by emphasizing the importance of patient-derived cell lines for investigating rare tumors and their progression.

## 5. Conclusions

In this study, we established two novel cell lines derived from a mucinous borderline ovarian tumor (MBOT). HMucBOT-1 preserved the morphological and molecular features of the original MBOT, whereas HMucBOT-2 displayed dedifferentiated characteristics, including features of mucinous and high-grade carcinoma. The differences between the two cell lines likely reflect the underlying tumor heterogeneity within MBOTs. One of the most important clinical implications of these findings is that the initial pathological diagnosis of surgical specimens does not always predict the clinical course, but unexpected relapse may occasionally occur. The clinicians should take note of this biology of MBOT.

Additionally, these cell lines serve as valuable models for studying the molecular basis of MBOT progression. HMucBOT-1 provides a platform for investigating early-stage tumor biology and preventive strategies, while HMucBOT-2 facilitates research into aggressive tumor behavior and malignant transformation. Together, these models support future efforts in molecular risk stratification, personalized therapy development, and improved clinical management of mucinous ovarian tumors.

## Figures and Tables

**Figure 1 cancers-17-01716-f001:**
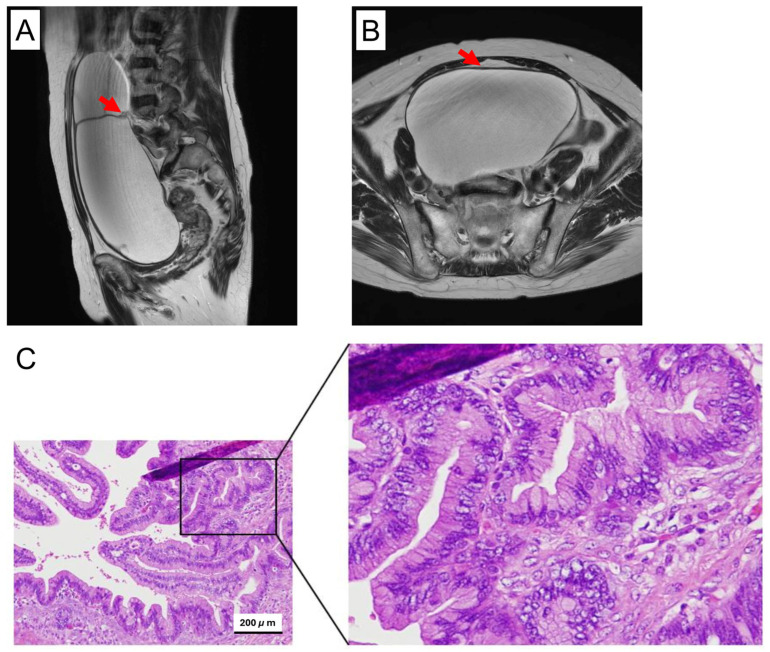
Histopathologic and morphologic features of the ovarian tumor. (**A**,**B**) Preoperative MR imaging of the original right ovarian tumor. The red arrow indicates thickened septa within the tumor. (**C**) Histological view of the HE-stained paraffin section of the ovarian tumor, representing a mucinous borderline tumor, exhibiting signs of mild atypia, such as an increased nucleus-to-cytoplasm ratio and prominent nucleoli, but lacking apparent interstitial invasion.

**Figure 2 cancers-17-01716-f002:**
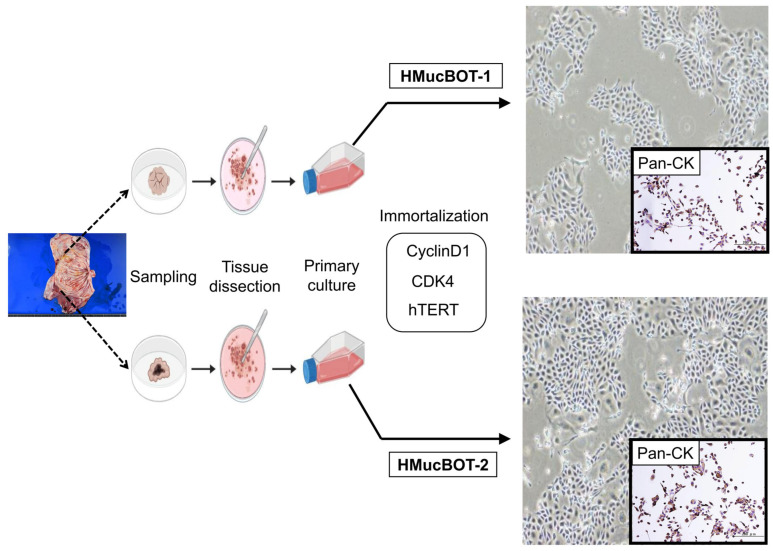
Primary culture from the mucinous ovarian tumor and the process of immortalization.

**Figure 3 cancers-17-01716-f003:**
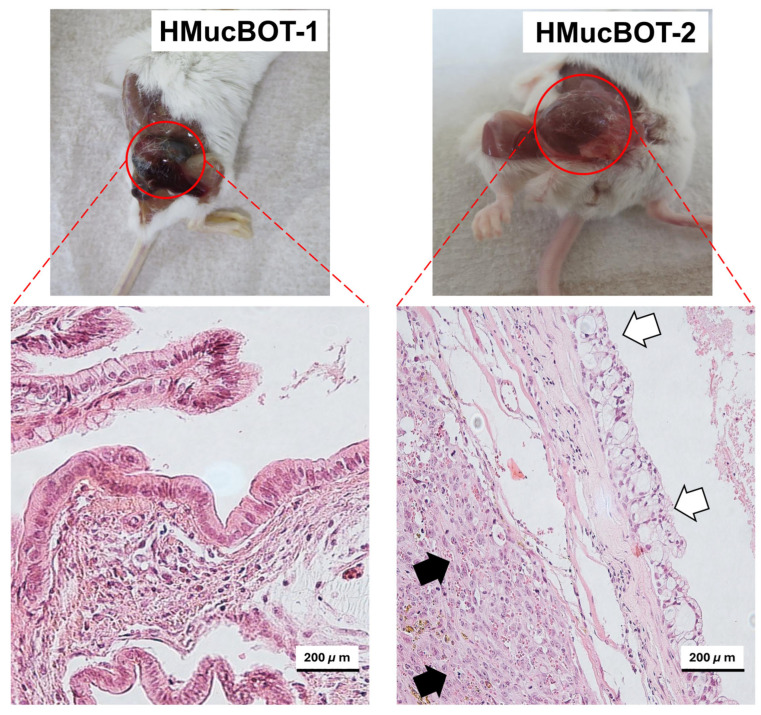
Mouse tumorigenicity to confirm histological reproduction of the original tumor.

**Figure 4 cancers-17-01716-f004:**
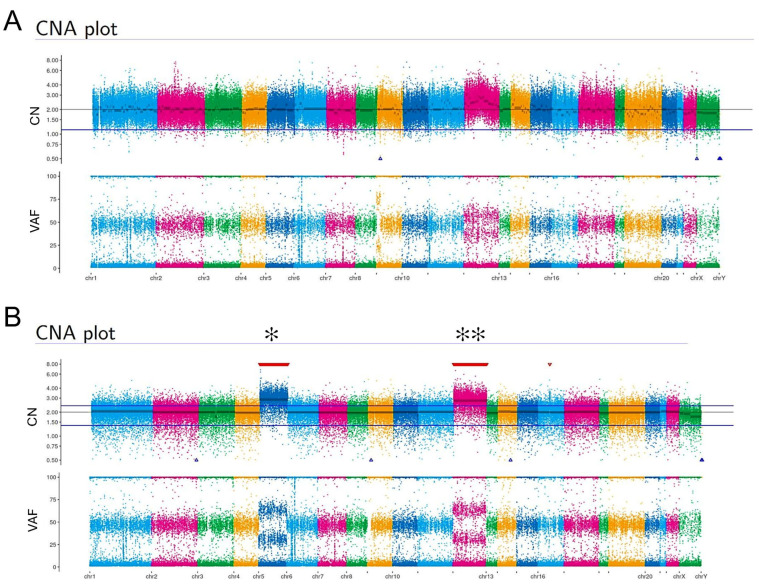
Copy number alterations of the original tumor and HMucBOT-2 cells. Copy number alterations (CNAs) and variant allele frequencies (VAFs) in the original ovarian tumor sample (**A**) and HMucBOT-2 cells (**B**). The significant chromosomal copy number alterations were identified in HMucBOT-2 cells at Chr 12 (shown as ✻✻), displaying a threefold gain in *KRAS*, *KMT2D*, *CLIP1*, and *CDK4*, and at Chr 5 (shown as ✻), with a threefold gain in *TERT*. CN, copy number.

**Figure 5 cancers-17-01716-f005:**
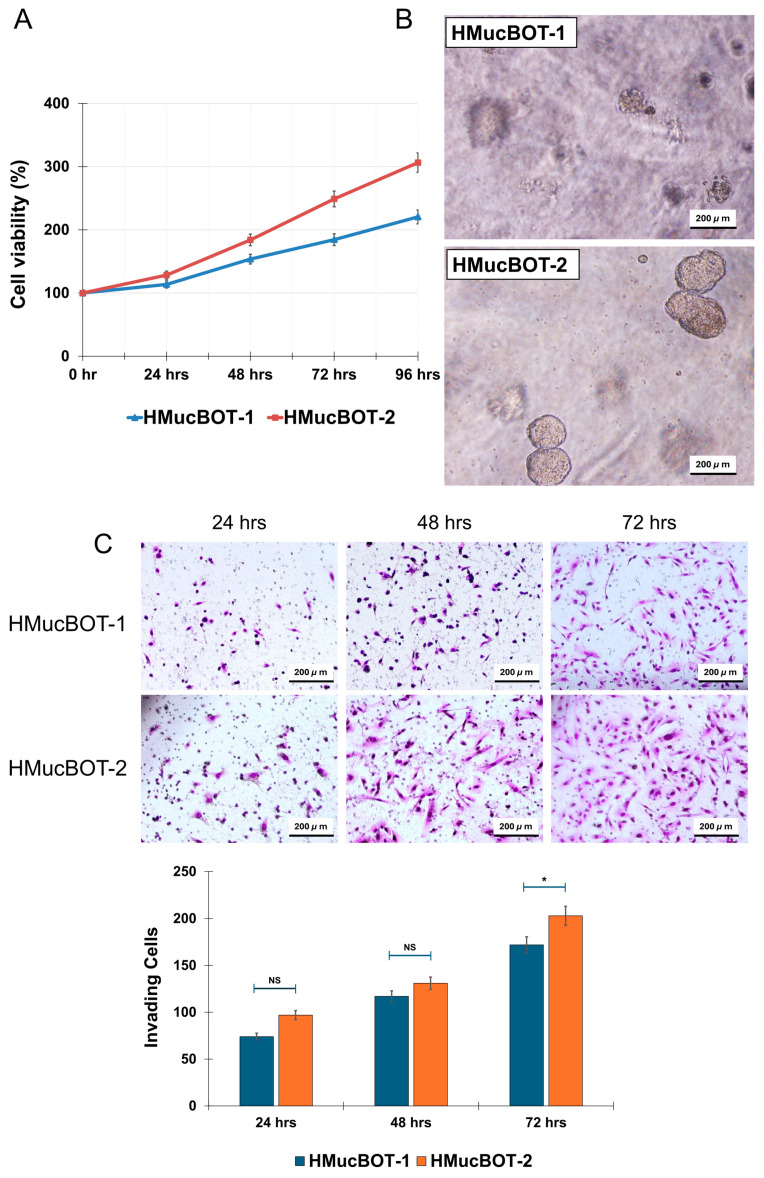
Growth properties of HMucBOT-1 and HMucBOT-2 cells. (**A**) MTT assay: The number of proliferating cells is shown for each cell type at a different time period. (**B**) Soft agar colony formation assay: Photographs of representative colonies (≥50 µm) after 14 days of seeding obtained through transmitted light microscopy. (**C**) Invasion assay: The number of invading cells is shown in each cell type at 24, 48, and 72 h. * *p* < 0.05. (**D**) Wound-healing assay: The number of migrating cells is shown in each cell type at different times. * *p* < 0.05. The error bar indicates the standard deviation (Student’s *t*-test was used; NS *p* > 0.05, * *p* < 0.05; replicates: n  =  3).

**Figure 6 cancers-17-01716-f006:**
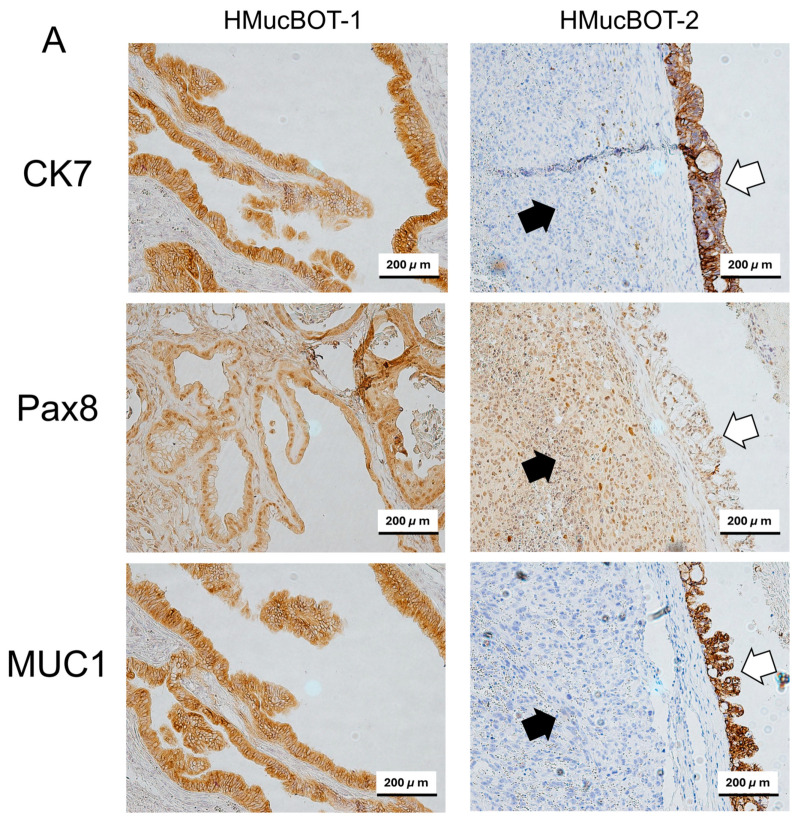
Immunohistochemical analysis of xenograft tumors of HMucBOT-1 and HMucBOT-2 cells. (**A**) Both tumors of HMucBOT-1 and HMucBOT-2 cells showed positive expression of CK7, PAX8, and MUC1. The blank arrows highlight areas of mucinous carcinoma, and the black arrows indicate regions of undifferentiated carcinoma. (**B**) Tumors of HMucBOT-2 cells showed expression of AE1/AE3, CAM5.2, and EMA in mucinous carcinoma lesions but not Desmin or S-100, with them exhibiting typical patterns of dedifferentiated carcinoma. (**C**) Tumors of HMucBOT-2 cells showed expression of Snail and Vimentin but not Twist in undifferentiated lesions. Mismatch repair proteins (MLH1, MLH6, MSH6, and PMS2) were all expressed in the tumor, in both mucinous carcinoma and undifferentiated lesions.

**Figure 7 cancers-17-01716-f007:**
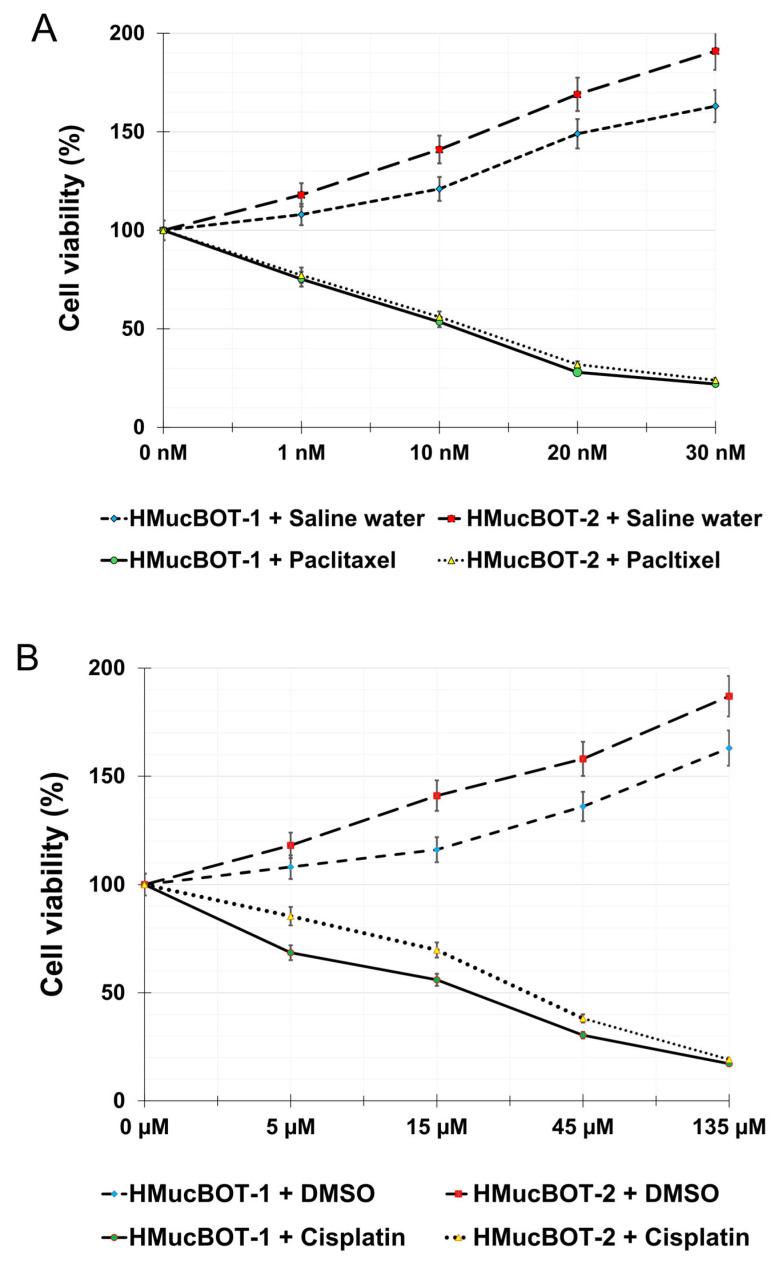
Chemosensitivity assay. (**A**) Cells were treated with paclitaxel (1, 10, 20, or 30 nmol/L) for 24, 48, or 72 h, together with a control (drug carrier). The IC50 value after 72 h of treatment was 11.66 nmol/L for HMucBOT-1 and 12.53 nmol/L for HMucBOT-2. (**B**) Cells were treated with cisplatin (5, 15, 45, or 135 µmol/L) for 24, 48, or 72 h, together with a control (drug carrier). The IC50 value after 72 h of treatment was 26.58 µmol/L for HMucBOT-1 and 28.55 µmol/L for HMucBOT-2.

**Table 1 cancers-17-01716-t001:** Tumor formation assay using HMucBOT-1 and HMucBOT-2 cell lines.

Cell Line Name	Results (Tumor Formation in SCID Mice)	Remarks
HMucBOT-1	5/5	Mucinous borderline tumor
HMucBOT-2	5/5	Dedifferentiated carcinoma

**Table 2 cancers-17-01716-t002:** Whole-exome sequencing analysis report of MBOT.

Gene Type	Gene	Mutation (VAF%)	Plession Score	Cos.	CV.	CNA	eCN
HRD	*ARID1A*	p.N106 * (21.6)	2			Neutral	2.5
HRD	*ARID1A*	p.L2270Ifs * 8 (23.0)	2			Neutral	2.5
OG	*KRAS*	p.G12V (22.3)	3	10,787	P	Neutral	2.8
TSG	*CDKN2A*	Wild type	3			HD	0.1
TSG	*ARID1B*	p.W1937 * (19.3)	2		P	Neutral	2.1
TSG	*NF1*	p.K354N (31.1)	2		P	Neutral	1.8
Other	*GAA*	p.S332L (12.1)	0.5			Neutral	……

MBOT, mucinous borderline ovarian tumor; HRD, homologous recombination deficiency; OG, oncogene; TSG, tumor suppressor gene; HD, homozygous deletion; P, pathogenic; VAF, variant allele frequency; Cos, cosmic; CV, ClinVar; CNA, copy number alteration; eCN, estimated copy number; *, indicating stop codon number.

**Table 3 cancers-17-01716-t003:** Whole-exome sequencing analysis report of HMucBOT-2.

Gene Type	Gene	Mutation (VAF%)	Plession Score	Cos.	CV.	CNA	eCN
HRD	*ARID1A*	p.N106 * (32.2)	2	0		Neutral	2.1
HRD	*ARID1A*	p.L2270Ifs * 8 (41.7)	2	0		Neutral	2.1
HRD	*BRIP1*	p.D153H (5.0)	1	0		Neutral	2.0
OG	*KRAS*	p.G12V (31.6)	3	10,787	P	Amp	3.1
OG	*ACSL3*	p.L186F (51.5)	1.5	2		Neutral	2
OG	*SUZ12*	p.V68A (6.2)	1	0		Neutral	2
OG	*TERT*	p.I587V (32.3)	1	0		Amp	3.3
OG	*GRM3*	p.R716M (6.9)	1	0		Neutral	2
OG	*EWSR1*	p.R511L (8.3)	1	0		Neutral	2
OG	*HMGA1*	p.K46R (5.9)	1	0		Neutral	2
OG	*IRF4*	p.A16T (54.0)	1	0		Neutral	2
OG	*GNAS*	p.S340A (50.1)	1	0		Neutral	2
OG	*EZH2*	p.C329S (6.6)	1	0		Neutral	2
OG	*BIRC6*	p.N2640K (5.4)	0.5	0		Neutral	2
OG	*MPL*	p.S162T (5.8)	0.5	0		Neutral	2.1
OG	*CLIP1*	p.K1156I (6.6)	0.5	0		Amp	3.1
OG	*PTK6*	p.G10V (6.7)	0.5	0		Neutral	2
OG	*SPECC1*	p.K950R (5.1)	0.5	0		Neutral	2
OG	*CCND1*	c.415-2A>T (22.6)	0.5	1		Neutral	2
OG	*PDE4DIP*	p.E1577G (6.7)	0.5	0		Neutral	2.1
OG	*CDK4*	c.633-1G>A (21.7)	0.5	0		Amp	3.1
OG	*CDK4*	c.219-2A>C (27.1)	0.5	0		Amp	3.1
OG	*CCND1*	c.414+1G>C (10.2)	0	0		Neutral	2
OG	*CCND1*	c.574+1G>A (8.6)	0	0		Neutral	2
OG	*CCND1*	c.724-1G>A (8.2)	0	0		Neutral	2
OG	*CDK4*	c.820-2A>T (9.6)	0	0		Amp	3.1
OG	*CDK4*	c.819+2T>A (10.7)	0	0		Amp	3.1
OG	*CDK4*	c.684-1G>A (6.4)	0	0		Amp	3.1
OG	*CDK4*	c.683+1G>C (8.9)	0	1		Amp	3.1
OG	*CDK4*	c.632+2T>C (13.9)	0	0		Amp	3.1
OG	*CDK4*	c.218+2T>C (10.4)	0	0		Amp	3.1
TSG	*ARID1B*	p.W1937(41.3)	2	0	P	Neutral	2
TSG	*NF1*	p.K354N (44.2)	2	0	P	Neutral	2
TSG	*ARID1B*	p.E1340(5.4)	1.5	0		Neutral	2
TSG	*ATP2B3*	c.2626-2A>G (5.2)	1.5	0		Neutral	1.7
TSG	*ARHGEF12*	p.Q1266L (6.5)	1	0		Neutral	2
TSG	*RAD21*	p.I17F (5.0)	1	1		Neutral	1.9
TSG	*PTPN13*	p.G2262V (5.8)	1	0		Neutral	1.9
TSG	*TPM3*	p.K6M (5.1)	1	0		Neutral	2.1
TSG	*ZFHX3*	p.M1294V (5.9)	1	0		Neutral	2
TSG	*SPEN*	p.Y2819N (6.5)	1	0		Neutral	2.1
TSG	*UBR5*	p.P2036H (6.1)	1	0		Neutral	1.9
TSG	*UBR5*	p.C2034G (5.0)	1	0		Neutral	1.9
TSG	*KMT2D*	p.A963T (5.5)	0.5	0		Amp	3.1
TSG	*CDKN2A*	Wild type	3			HD	−0.7
Other	*APOB*	p.E1601K (43.5)	1	0			-
Other	*GAA*	p.S332L (44.3)	1	0			-
Other	*POLQ*	p.L2352F (5.7)	0.5	0		Neutral	2
Other	*PRKCB*	p.R415S (6.0)	0.5	0		Neutral	2
Other	*NBEA*	c.6585+2958G>C (6.4)	0.5	0		Neutral	1.9
Other	*TTN*	p.T25595P (5.0)	0.5	0			-
Other	*TTN*	p.V21151L (5.0)	0.5	0			-
Other	*CTNND1*	p.P88T (5.2)	0.5	0		Neutral	2
Other	*PCBP1*	p.N48H (6.1)	0.5	0		Neutral	2
Other	*TMEM43*	p.G237E (6.9)	0.5	0			-
Other	*AFDN*	p.S1469R (6.2)	0.5	0			-
Other	*TTN*	p.I21153F (5.2)	0.5	0			-
Other	*TTN*	c.11311+3989T>G (5.3)	0.5	0			-

HRD, homologous recombination deficiency; OG, oncogene; TSG, tumor suppressor gene; HD, homozygous deletion; P, pathogenic; Amp, amplification; VAF, variant allele frequency; Cos, cosmic; CV, ClinVar; CNA, copy number alteration; eCN, estimated copy number; *, indicating stop codon number.

## Data Availability

The data presented in this study are available on request from the corresponding authors.

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
