# Peer review of "Establishment of Two Novel Ovarian Tumor Cell Lines with Characteristics of Mucinous Borderline Tumors or Dedifferentiated Carcinoma—Implications for Tumor Heterogeneity and the Complex Carcinogenesis of Mucinous Tumors"

_cancers, 2025, doi:10.3390/cancers17101716_

Round 1
Reviewer 1 Report
Comments and Suggestions for Authors
The study provides valuable insights into tumor heterogeneity and the molecular mechanisms underlying MBOT progression. The research is methodologically sound, with comprehensive analyses including histopathology, genetic profiling, and functional assays. However, there are minor grammatical and typographical issues, and the discussion could benefit from additional references to contextualize the findings more broadly.
Abstract:
Line 10: "Their molecular carcinogenesis and tumor biology remain largely unknown, and experimental models of this tumor are extremely rare." Consider rephrasing for clarity, e.g., "The molecular mechanisms driving their carcinogenesis and tumor biology remain poorly understood, and experimental models for this tumor type are scarce."
Line 15: "In a patient-derived xenograft model, one line retained the original morphological characteristics of the MBOT; in comparison, the other line displayed a transition to mucinous carcinoma accompanying undifferentiated carcinoma..." Rephrase for conciseness, e.g., "In a patient-derived xenograft model, HMucBOT-1 retained MBOT morphology, whereas HMucBOT-2 exhibited features of mucinous carcinoma with undifferentiated components."
Introduction:
Line 2: Update the statistics on overall cancer incidence and the prevalence of this specific cancer type, including survival rates, to emphasize the urgent need for cancer studies. Cite Cancer Statistics, 2024. Additionally, provide a general overview of cancer therapy, referencing the NIH paper“Cancer treatments: Past, present, and future, 2024” for further insights.
Line 5: "Borderline ovarian tumors (BOTs) are fundamentally characterized by intermediate clinical aggressiveness with pathological features of moderate cellular proliferation and slight nuclear atypia but lacking destructive stromal invasion [1, 2]." Cite additional recent reviews to strengthen the context of BOT classification and clinical significance.
Line 15: "MBOTs, also known as atypical proliferative mucinous tumors, feature mucin-producing cells with borderline malignant potential [7–9]..." Include references to studies discussing the clinical outcomes of MBOTs to underscore their significance.
Line 25: "The most representative mutations detected in mucinous tumors are those in the mitogen-activated protein kinase (MAPK) pathway, including KRAS mutations and ERBB2 amplification/overexpression [12]." Expand on the functional implications of these mutations in MBOT progression.
Methods and Materials:
Line 8: "Tissue samples were collected and submitted to primary culture based on previously established protocols [14,15]." Clarify the specific modifications or optimizations made to these protocols for MBOT-derived cells.
Line 20: "The resultant cells were designated HMucBOT-1 and HMucBOT-2, cultured in a mixture of DMEM and F12 (Company, location, etc.) at a ratio of 1:3 (WAKO, Osaka, Japan)..." Specify the rationale for the 1:3 ratio and its relevance to MBOT cell culture.
Line 35: "The samples were then submitted for STR profiling analysis (BEX CO., LTD, Japan) targeting 16 loci (Supplementary Table 1)." Include a citation or validation source for the STR profiling method.
Results:
Line 5: "The HMucBOT-2 cells showed much greater proliferation capacity compared to the HMucBOT-1 cells, with the doubling time for the HMucBOT-2 cells being 58 hours compared to 87 hours for the HMucBOT-1 cells (Figure 5A)." Cite studies linking doubling time to tumor aggressiveness in ovarian cancer models.
Line 15: "These findings highlight how HMucBOT-2 cells exhibit more aggressive and invasive features than HMucBOT-1 cells." Discuss potential molecular mechanisms driving this phenotypic divergence, supported by literature.
Discussion:
Line 5: "We speculate that there are two possible explanations for this diversity: either cells with borderline features in the sampling tissues progressively developed into carcinoma during culture or tumorgenicity in the mice or cells with carcinoma features were already present in the sampled tissues but only in minute portions that could not be detected through routine pathological inspection." Rephrase for clarity, e.g., "We propose two hypotheses: (1) borderline cells acquired carcinoma traits during culture or xenografting, or (2) carcinoma cells were present but undetected in the original tumor."
Line 20: "The whole-exome sequences of the original tumor and HMucBOT-2 cells revealed a variety of common oncogenic mutations, such as KRAS, ARID1A, ARID1B, SMO, BRIP1, and NF1, while maintaining a wild-type status for the CDKN2A gene." Compare these findings with other studies on MBOT genetic profiles to highlight novelty or consistency. Discussion bias in the sequencing methodology, cite paper such as “Technical and Biological Biases in Bulk Transcriptomic Data Mining for Cancer Research, 2025”
Line 30: "Copy number changes are the key drivers that influence the grade and progression of metastatic disease, contributing to uncontrolled cell growth, increased invasion, and therapy resistance [38–40]." Give a few other example of Copy number change in cancer, cite paper such as “The pan-cancer landscape of glutamate and glutamine metabolism: A comprehensive bioinformatic analysis across 32 solid cancer types,2024”. Expand on how these alterations specifically impact MBOT biology. Recent studies have highlighted advancements in liquid biopsies for cancer diagnostics and monitoring. Research such as “Updates on liquid biopsies in neuroblastoma for treatment response, relapse and recurrence assessment, 2024”demonstrates the utility of circulating tumor DNA (ctDNA) detection through liquid biopsy techniques. Additionally, emerging sequencing technologies have improved the sensitivity and specificity of DNA analysis, such as “Development of a molecular barcode detection system for pancreaticobiliary malignancies and comparison with next-generation sequencing, 2024”. Also the methylation is also used for detection, reported in “Methylation signatures as biomarkers for non-invasive early detection of breast cancer: A systematic review of the literature, 2024”. Please cited these related papers and discuss: consider whether the mechanisms discussed in this study could be identified through these diagnosis methods.
Discuss the establishment of difference cell model for tumor heterogeneity, cite paper such as “Establishment of Breast Phyllodes Tumor Cell Lines Preserving the Features of Phyllodes Tumors”
Comments on the Quality of English Languageok
Author Response
Thank you very much for your valuable comments and constructive feedback. We appreciate your recognition of the strengths of our study in addressing tumor heterogeneity and the molecular mechanisms of MBOT progression.
In response to your suggestions, we have carefully reviewed the manuscript and corrected minor grammatical and typographical errors throughout the text. Additionally, we have expanded the discussion section by incorporating more relevant references to improve our manuscript quality.
We hope these revisions meet your expectations and further enhance the clarity and scientific value of our work.
Abstract:
Comment 1: "Their molecular carcinogenesis and tumor biology remain largely unknown, and experimental models of this tumor are extremely rare." Consider rephrasing for clarity, e.g., "The molecular mechanisms driving their carcinogenesis and tumor biology remain poorly understood, and experimental models for this tumor type are scarce."
Response 1: Thank you for your helpful suggestion. We agree that rephrasing the sentence improves clarity. Accordingly, we have revised the sentence as “The molecular mechanisms driving their carcinogenesis and tumor biology remain poorly understood, and experimental models for this tumor type are scarce. The revisions are highlighted in the manuscript and viewable with the track changes feature.
Comment 2: In a patient-derived xenograft model, one line retained the original morphological characteristics of the MBOT; in comparison, the other line displayed a transition to mucinous carcinoma accompanying undifferentiated carcinoma..." Rephrase for conciseness, e.g., "In a patient-derived xenograft model, HMucBOT-1 retained MBOT morphology, whereas HMucBOT-2 exhibited features of mucinous carcinoma with undifferentiated components."
Response 2: Thank you for your valuable suggestion. We agree that rephrasing the sentence improves clarity. Accordingly, we have updated and are viewable with the track changes feature to better reflect your recommendation.
Introduction:
Comment 1: Update the statistics on overall cancer incidence and the prevalence of this specific cancer type, including survival rates, to emphasize the urgent need for cancer studies. Cite Cancer Statistics, 2024. Additionally, provide a general overview of cancer therapy, referencing the NIH paper“Cancer treatments: Past, present, and future, 2024” for further insights.
Response 1: Thank you for your insightful suggestion. In response, we have updated the manuscript to include the most recent statistics on overall cancer incidence and survival rates, as reflected in citations [1–4] and indicated through track changes. Additionally, we have incorporated a general overview of current and emerging cancer therapies by referencing the NIH review article Cancer Treatments: Past, Present, and Future, 2024 (citation [24]). These additions provide a broader context for our study and highlight the continued need for innovative strategies to improve cancer care and patient outcomes.
Comment 2: Borderline ovarian tumors (BOTs) are fundamentally characterized by intermediate clinical aggressiveness with pathological features of moderate cellular proliferation and slight nuclear atypia but lacking destructive stromal invasion [1, 2]." Cite additional recent reviews to strengthen the context of BOT classification and clinical significance.
Response 2: We sincerely appreciate your valuable suggestion. In response, we have included an additional recent review, as recommended, to further strengthen the context of BOT classification and clinical significance. This has been reflected in citation [7] in the revised manuscript.
Comment 3: "MBOTs, also known as atypical proliferative mucinous tumors, feature mucin-producing cells with borderline malignant potential [7–9]..." Include references to studies discussing the clinical outcomes of MBOTs to underscore their significance.
Response 3: Thank you for your valuable comment. In response, we have included additional references (citations [15–20]) that discuss the clinical outcomes of MBOTs to better emphasize their clinical significance in the revised manuscript.
Comment 4: The most representative mutations detected in mucinous tumors are those in the mitogen-activated protein kinase (MAPK) pathway, including KRAS mutations and ERBB2 amplification/overexpression [12]." Expand on the functional implications of these mutations in MBOT progression.
Response 4: We are grateful for your constructive suggestion. To address your suggestion, we have added an additional reference (citation [24]) that elaborates on the functional implications of these mutations in the progression of MBOT. We hope this addition enhances the depth and relevance of our discussion.
Methods and Materials:
Comment 1: Tissue samples were collected and submitted to primary culture based on previously established protocols [14,15]." Clarify the specific modifications or optimizations made to these protocols for MBOT-derived cells.
Response 1: Thank you for your valuable inquiry. We used our previously established protocol tailored for primary epithelial cell isolation from ovarian tumors. As the procedure was confirmed to be well-suited for a variety of types of epithelial tissues, including MBOT-derived tissues, no further modifications or optimizations were necessary for this study.
Comment 2: "The resultant cells were designated HMucBOT-1 and HMucBOT-2, cultured in a mixture of DMEM and F12 (Company, location, etc.) at a ratio of 1:3 (WAKO, Osaka, Japan)..." Specify the rationale for the 1:3 ratio and its relevance to MBOT cell culture.
Response 2: Thank you for your thoughtful comment and attention to detail. In response, we have included the company and location information as suggested. The DMEM:F12 (1:3) ratio was chosen based on established literature in ovarian epithelial cell culture (citation [28]), as it offers a balanced nutrient environment that supports optimal epithelial cell proliferation.
Comment 3: "The samples were then submitted for STR profiling analysis (BEX CO., LTD, Japan) targeting 16 loci (Supplementary Table 1)." Include a citation or validation source for the STR profiling method.
Response 3: Thank you very much for your valuable suggestion. In response, we have added a citation to support the validation of the STR profiling method used in our study (citation [31]). We believe this addition improves the methodological transparency and further reinforces the reliability of our approach.
Results:
Comment 1: "The HMucBOT-2 cells showed much greater proliferation capacity compared to the HMucBOT-1 cells, with the doubling time for the HMucBOT-2 cells being 58 hours compared to 87 hours for the HMucBOT-1 cells (Figure 5A)." Cite studies linking doubling time to tumor aggressiveness in ovarian cancer models.
Response 1: Thank you for your thoughtful comment. In response, we have added relevant citations in the Results section (Section 3.5; citations [36, 37]) to support the association between cell doubling time and tumor aggressiveness in ovarian cancer models. We believe this addition strengthens the interpretation of our proliferation data and enhances the scientific context of our findings.
Comment 2: "These findings highlight how HMucBOT-2 cells exhibit more aggressive and invasive features than HMucBOT-1 cells." Discuss potential molecular mechanisms driving this phenotypic divergence, supported by literature.
Response 2: We sincerely thank you for your insightful suggestion. In response, we have expanded the discussion in Results Section 3.5 to explore potential molecular mechanisms underlying the phenotypic divergence between HMucBOT-1 and HMucBOT-2 cells, supported by relevant literature (citations [38–40]). We believe this addition enriches the interpretation of our findings and offers a more comprehensive understanding of the observed cellular characteristics.
Discussion:
Comment 1: "We speculate that there are two possible explanations for this diversity: either cells with borderline features in the sampling tissues progressively developed into carcinoma during culture or tumorgenicity in the mice or cells with carcinoma features were already present in the sampled tissues but only in minute portions that could not be detected through routine pathological inspection." Rephrase for clarity, e.g., "We propose two hypotheses: (1) borderline cells acquired carcinoma traits during culture or xenografting, or (2) carcinoma cells were present but undetected in the original tumor."
Response 1: Thank you for your helpful suggestion. We have slightly rephrased the paragraph to improve clarity, although not exactly as proposed. We hope the updated wording effectively conveys the intended meaning while maintaining consistency with the overall tone of the manuscript.
Comment 2: "The whole-exome sequences of the original tumor and HMucBOT-2 cells revealed a variety of common oncogenic mutations, such as KRAS, ARID1A, ARID1B, SMO, BRIP1, and NF1, while maintaining a wild-type status for the CDKN2A gene." Compare these findings with other studies on MBOT genetic profiles to highlight novelty or consistency. Discussion bias in the sequencing methodology, cite paper such as “Technical and Biological Biases in Bulk Transcriptomic Data Mining for Cancer Research, 2025”
Response 2: Thank you very much for your valuable suggestion. In response, we have expanded the discussion to compare our genetic findings with those reported in previous studies on MBOT, highlighting both the consistencies and potential novel aspects of our results (citations [59, 60]). Additionally, we have addressed potential biases associated with the sequencing methodology and included a reference to the recent article “Technical and Biological Biases in Bulk Transcriptomic Data Mining for Cancer Research, 2025” (citation [58]) to acknowledge these limitations and enhance the transparency of our analysis.
Comment 3: "Copy number changes are the key drivers that influence the grade and progression of metastatic disease, contributing to uncontrolled cell growth, increased invasion, and therapy resistance [38–40]." Give a few other example of Copy number change in cancer, cite paper such as “The pan-cancer landscape of glutamate and glutamine metabolism: A comprehensive bioinformatic analysis across 32 solid cancer types,2024”. Expand on how these alterations specifically impact MBOT biology. Recent studies have highlighted advancements in liquid biopsies for cancer diagnostics and monitoring. Research such as “Updates on liquid biopsies in neuroblastoma for treatment response, relapse and recurrence assessment, 2024”demonstrates the utility of circulating tumor DNA (ctDNA) detection through liquid biopsy techniques. Additionally, emerging sequencing technologies have improved the sensitivity and specificity of DNA analysis, such as “Development of a molecular barcode detection system for pancreaticobiliary malignancies and comparison with next-generation sequencing, 2024”. Also the methylation is also used for detection, reported in “Methylation signatures as biomarkers for non-invasive early detection of breast cancer: A systematic review of the literature, 2024”. Please cited these related papers and discuss: consider whether the mechanisms discussed in this study could be identified through these diagnosis methods.
Discuss the establishment of difference cell model for tumor heterogeneity, cite paper such as “Establishment of Breast Phyllodes Tumor Cell Lines Preserving the Features of Phyllodes Tumors”
Response 3: We sincerely appreciate your helpful advice. In response, we have revised the manuscript to include additional examples of copy number alterations in cancer and also discussed the potential impact of such alterations on MBOT biology (citation [80-89]).
Furthermore, we incorporated recent advances in diagnostic approaches, including circulating tumor DNA (ctDNA), methylation-based detection, and molecular barcoding, with references to the following studies: “Updates on liquid biopsies in neuroblastoma...” (citation [90]), “Methylation signatures as biomarkers...” (citation [91]), and “Development of a molecular barcode detection system...” (citation [92]).
Lastly, to address tumor heterogeneity, we included a reference to the study “Establishment of Breast Phyllodes Tumor Cell Lines...” (citation [93]). These additions strengthen our discussion and provide a broader clinical and research context for our findings.
Final Notes:
We are grateful for your thoughtful critique, which helped us improve both the scientific content and clarity of our manuscript. All suggested references and revisions have been carefully incorporated.
Reviewer 2 Report
Comments and Suggestions for Authors
In this study two cell lines from a single patient with mBOT were established in order to evaluate their biological features. It is a well-written manuscript with clearly presented methodology and results. I have the following comments:
- What percentage of mBOT recur as mucinous carcinoma? This information should be added to the introduction section.
- In the conclusions section the authors should further discuss a possible clinical implications of this study.
Author Response
We sincerely thank the reviewer for their kind and positive encouraging comments on our study. We are grateful for the recognition of the significance of establishing two MBOT-derived cell lines and their potential value in future research. We truly appreciate the reviewer’s acknowledgment of the relevance and contribution of our work. Our responses are listed below-
Comment 1: What percentage of mBOT recur as mucinous carcinoma? This information should be added to the introduction section.
Response 1: Thank you very much for your valuable suggestion. We agree that including information on the recurrence rate of MBOTs progressing to mucinous carcinoma adds important context to the background. In response, we have incorporated this information into the introduction section, along with relevant supporting references (citations [15–20]) to enhance the clarity and significance of our study.
Comment 2: “In the conclusions section the authors should further discuss a possible clinical implications of this study.”
Response 2: Thank you for your thoughtful comment. In response, we have expanded the conclusion section to further discuss the potential clinical implications of our findings. We believe this addition strengthens the relevance of our study by highlighting how the established cell lines may contribute to future diagnostic, therapeutic, and translational research efforts in the context of mucinous ovarian tumors.
Final Note: We would like to express our sincere gratitude to the reviewer for thoughtful and constructive comments. Your valuable insights have greatly contributed to enhancing the quality and clarity of our manuscript.
Reviewer 3 Report
Comments and Suggestions for Authors
cancers-two Mbot cell lines
Title: Establishment of two novel ovarian tumor cell lines with char-2 acteristics of mucinous borderline tumors or dedifferentiated 3 carcinoma—Implications for tumor heterogeneity and the com-4 plex carcinogenesis of mucinous tumors
Summary
The author established two Mbot cell lines for future research. The cell lines could retain the characteristics of the parental tumor.
Comment
- MBOT is a rare type of ovarian tumor. The established cell lines were valuable for tumor heterogeneity studies.
- The reason why you choose cyclin D1 and CDK4 should be illustrated. Usually, we use hTERT and large t antigen to immortalize the cell line. Whether you selected cyclin D1 and CDK4 influence the gene expression?
- The abbreviation in Table 2 and 3 should be explained. HRD, OG, TSG, amp, HD
- The meaning of CNA and VAF could be explained.
- Could a gene mutation in MBOT-2 explain the biological phenomenon of undifferentiation carcinoma? Could the difference between BOT and carcinoma be explained?
- What is the clinical outcome of the MBOT-2 patient? Will these findings change the therapy for the patient?
Author Response
We sincerely thank the reviewer for the kind and encouraging comments regarding our study. We are grateful for the recognition of the significance of establishing two MBOT-derived cell lines and their potential utility in future research. We truly appreciate the reviewer’s acknowledgment of the relevance of our work. We have carefully addressed each comment, as outlined in the responses below.
Comment 1: MBOT is a rare type of ovarian tumor. The established cell lines were valuable for tumor heterogeneity studies.
Response 1: Thank you for recognizing the importance of our work. Indeed, MBOTs are rare, and the successful establishment of two distinct cell lines provides a unique opportunity to investigate the tumor heterogeneity and malignant transformation potential inherent in MBOTs.
Comment 2: The reason why you choose Cyclin D1 and CDK4 should be illustrated. Usually, we use hTERT and large T antigens to immortalize the cell line. Whether you selected cyclin D1 and CDK4 influence to gene expression?
Response 2: Thank you for this important question. In our study, we selected Cyclin D1 and CDK4 in combination with hTERT to immortalize the epithelial cells. It is well known that primary epithelial cells typically have a limited lifespan in culture due to two major senescence barriers: early stress-induced senescence mediated by the Rb pathway and later replicative senescence caused by telomere shortening.
Based on previous evidence, the Cyclin D1/CDK4 approach effectively bypasses the initial senescence barrier by inactivating the Rb pathway, while hTERT expression restores telomerase activity, thereby overcoming replicative senescence. Importantly, we avoided the use of viral oncogenes such as SV40 large T antigen, because it can disrupt not only p53 but also other tumor suppressor functions, leading to undesirable transformation. Most importantly, SV40LT triggers severe genetic instability. Therefore, it is not suitable to establish immortal cells with physiologically normal functional characteristics.
Another study demonstrated that the combinatorial expression of cell cycle regulators is a more suitable and physiologically relevant strategy for epithelial cell immortalization compared to oncogenic methods and they proved that K4DT is the more reliable method among these three methods (SV40, E6/E7, or K4DT) (DOI: 10.1016/j.isci.2020.101929). This approach allows for long-term proliferation while preserving cellular morphology, gene expression profiles, and biological characteristics. In particular, Cyclin D1/CDK4 + hTERT immortalization has been shown to maintain normal karyotypes and avoid full transformation, making it highly appropriate for studies focused on ovarian epithelial tumor biology [Sasaki et al., 2009, https://doi.org/10.1093/carcin/bgp007; Shiomi et al. doi:10.1038/gt.2011.44].
Comment 3: The abbreviation in Table 2 and 3 should be explained. HRD, OG, TGS, amp, HD.
Response 3: Thank you for pointing this out. We have now included full definitions of all abbreviations used in Tables 2 and 3 within the figure legends and added a footnote for clarity.
Comment 4: The meaning of CNA and VAF could be explained.
Response 4: We sincerely appreciate the reviewer’s thoughtful suggestion. In response, we have provided the full forms of the abbreviations CNA (copy number alteration) and VAF (variant allele frequency) at their first mention in the manuscript, as well as in the table and figure legends where they were previously undefined.
Comment 5: Could a gene mutation in HMucBOT-2 explain the high biological phenomenon of undifferentiated carcinoma? Could the difference between BOT and carcinoma be explained?
Response 5: Thank you for your insightful question and this is indeed a critical point. Our analysis revealed that HMucBOT-2 harbors additional genomic alterations that do not present in the parental tumor or HMucBOT-1, including copy number alterations in mutant KRAS, as well as KMT2D and CLIP1 gene amplification. These molecular changes may contribute to the transition toward an undifferentiated phenotype.
To further validate this biological phenomenon, a potential approach would be to introduce the specific genetic alterations identified in HMucBOT-2 into HMucBOT-1 cells with MBOT characteristics. Subsequent xenografting of these modified cells into mice could help determine whether these alterations are sufficient to drive the development of undifferentiated carcinoma. This experiment would provide more direct evidence supporting the role of these genomic events in malignant transformation, which is currently ongoing in our laboratory as an independent study.
Comment 6: What is the clinical outcome of the MBOT-2 patient? Will these findings change the therapy for the patient?
Response 6: We appreciate the reviewer’s interest in the clinical correlation. As this study is based on a single patient diagnosed with MBOT, from whom both HMucBOT-1 and HMucBOT-2 cell lines were established, broader clinical conclusions are limited. The patient has been closely monitored postoperatively and remains disease-free, with no evidence of recurrence to date. While our findings do not directly influence the patient's treatment, the identification of an aggressive cellular clone (HMucBOT-2) from the same tumor highlights the importance of vigilant clinical follow-up in MBOT cases. These results also support the potential utility of molecular profiling for risk stratification and individualized surveillance strategies in future clinical practice.
Final Note: Once again, we extend our sincere thanks to the reviewer for the thoughtful feedback and appreciation of our study’s significance.
Round 2
Reviewer 1 Report
Comments and Suggestions for Authors
ok
Comments on the Quality of English Languagegood